# Passive exposure to task-relevant stimuli enhances categorization learning

**Christian Schmid**[1†]**, Muhammad Haziq**[1†]**, Melissa M Baese-Berk**[2]**, James M Murray**[1*‡]**, Santiago Jaramillo**[1*‡]

[1]Institute of Neuroscience, University of Oregon, Eugene, United States; [2]Department of Linguistics, University of Oregon, Eugene, United States

**\*For correspondence:**
jmurray9@uoregon.edu (JMM);
sjara@uoregon.edu (SJ)

[†]These authors contributed equally to this work

[‡]Co-senior authors

**Competing interest:** The authors declare that no competing interests exist.

**Abstract** Learning to perform a perceptual decision task is generally achieved through sessions of effortful practice with feedback. Here, we investigated how passive exposure to task-relevant stimuli, which is relatively effortless and does not require feedback, influences active learning. First, we trained mice in a sound-categorization task with various schedules combining passive exposure and active training. Mice that received passive exposure exhibited faster learning, regardless of whether this exposure occurred entirely before active training or was interleaved between active sessions. We next trained neural-network models with different architectures and learning rules to perform the task. Networks that use the statistical properties of stimuli to enhance separability of the data via unsupervised learning during passive exposure provided the best account of the behavioral observations. We further found that, during interleaved schedules, there is an increased alignment between weight updates from passive exposure and active training, such that a few interleaved sessions can be as effective as schedules with long periods of passive exposure before active training, consistent with our behavioral observations. These results provide key insights for the design of efficient training schedules that combine active learning and passive exposure in both natural and artificial systems.

## eLife assessment

This study reports **valuable** behavioral and computational observations regarding how passive exposure to auditory stimuli can facilitate auditory categorization. The combination of behavioral results in mice with a study of artificial neural network models provides **solid** evidence for the authors' conclusions. This paper will likely be of broad interest to the general neuroscience community.

## Introduction

Active learning of a perceptual decision task requires both expending effort to perform the task and having access to feedback about task performance. Passive exposure to sensory stimuli, on the other hand, is relatively effortless and does not require feedback about performance. Since animals are continuously exposed to stimuli in their environment, the nervous system could take advantage of this passive exposure, for example by learning features related to the statistical structure of the stimulus distribution, to increase the speed and efficiency of active task learning. For auditory learning in particular, schedules that effectively combine active training and passive exposure could yield more efficient approaches for learning to discriminate ethologically relevant sounds (as needed for example during second-language learning or musical training in humans) compared to active training alone.

A large body of research has demonstrated that exposure to sounds early in life influences the ability to discriminate acoustic stimuli (*Kuhl et al., 2003*; *Maye et al., 2002*; *Kral, 2013*). However, the conditions under which passive exposure in the adult can help auditory learning are not well

understood. In humans, previous work has demonstrated that, under specific conditions, interleaved passive exposure is beneficial for learning, sometimes to the extent that active sessions can be replaced with passive exposure and still yield similar performance (*Wright et al., 2015*). In other animals, which provide greater experimental access for investigating the neural mechanisms of learning, studies have focused mostly on the effects of perceptual learning (the experience-dependent enhancement in sensory discrimination) from exposure to stimuli during active training (*Bao et al., 2004*; *Polley et al., 2006*; *Caras and Sanes, 2017*). Although some progress has been made for other sensory modalities, such as olfaction (*Fleming et al., 2019*), the question of whether and how the combination of passive exposure with active training improves auditory learning in animal models has received little attention, limiting the ability to investigate the neural mechanisms that might be involved.

Using inexpensive unlabeled data during passive exposure to improve the efficiency of active training is also of great interest for machine learning, where large quantities of labeled training data are not always readily available. Recent approaches for training deep networks for speech recognition have successfully used large quantities of unlabeled data to achieve state-of-the-art levels of performance with minimal active training (*Baevski et al., 2020*; *Baevski et al., 2021*). Moreover, theoretical work inspired by neurobiology has argued that unsupervised learning, which may occur during passive exposure, could modify neural representations in such a way as to later facilitate more efficient supervised learning (*Nassar et al., 2021*). However, optimal ways to combine supervised and unsupervised learning, as well as the mechanisms that may underlie such benefits from passive exposure, remain unknown.

As a first step toward addressing these gaps in knowledge, we evaluated whether passive exposure to sounds improves learning of a sound-categorization task in mice. We found that passive presentation of stimuli enhanced learning speed in mice, either when passive presentation occurred before any active training or when passive-exposure sessions were interleaved with active training. Then, we performed a theoretical analysis of learning in artificial neural networks that combine different learning rules to identify the conditions under which they account for the experimental data. Our theoretical analysis indicates that the experimentally observed benefits of passive exposure can be accounted for by networks in which unsupervised learning in early layers shapes neural representations of sensory stimuli, while supervised learning in later layers uses these representations to drive behavior.

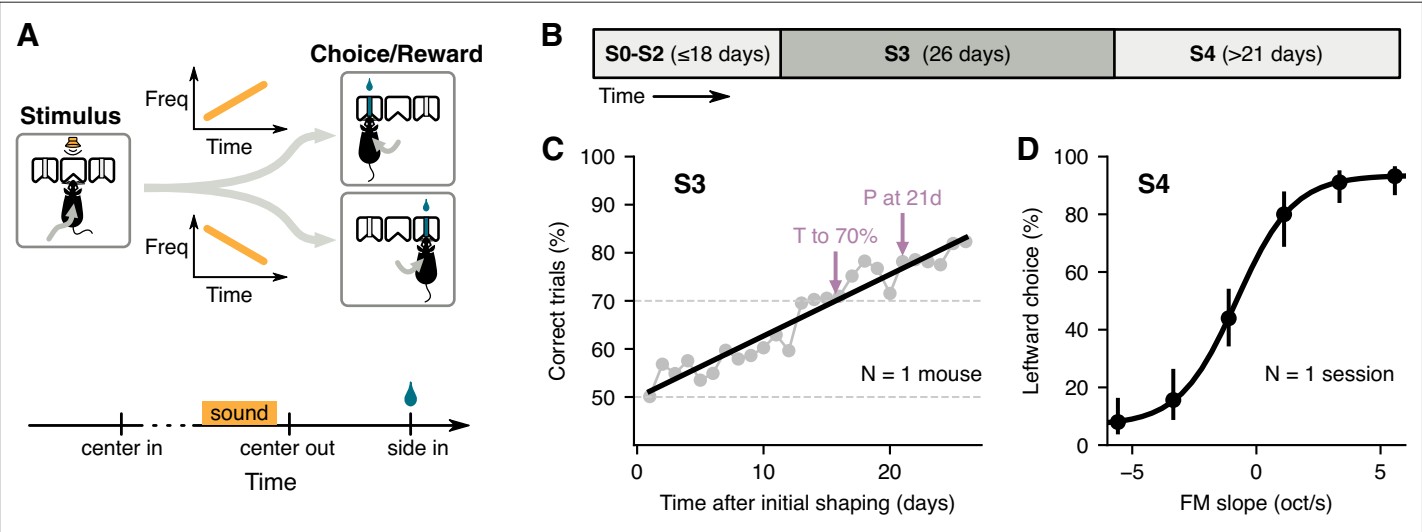

**Figure 1.** Two-alternative choice sound-categorization task for mice. (**A**) Mice initiated a trial by poking their nose into the center port of a three-port chamber, triggering the presentation of a frequency-modulated (FM) sound. To obtain reward, animals had to choose the correct side port according to the slope of the frequency modulation (left for positive slopes, right for negative slopes). (**B**) Training schedule: mice underwent several shaping stages (S0–S2) before learning the full task; the main learning stage (stage S3) used only the highest and lowest FM slopes; psychometric performance was evaluated using 6 different FM slopes (stage S4). (**C**) Daily performance for one mouse during S3. Arrows indicate estimates of the time to reach 70% correct and the performance at 21 days given a linear fit (black line). (**D**) Average leftward choices for each FM slope during one session of S4 for the mouse in C. Error bars indicate 95% confidence intervals.

## Results

### Learning a sound-categorization task

We first designed a two-alternative choice sound-categorization task to allow testing the effects of passive exposure on categorization learning. In this task, freely moving mice had to discriminate whether the slope of a frequency-modulated sound was positive or negative. Animals initiated each trial by poking the center port of a three-port chamber, at which point a 200-ms sound was presented after a brief silent delay. Mice then had to choose the left or right reward ports depending on the slope of the stimulus (*Figure 1A*). Animals were trained once per day using a schedule with the following stages (*Figure 1B*): shaping stages (S0–S2), in which animals learned to poke and obtain water reward; the main training stage with two stimuli (S3), in which animals learned to associate a stimulus with a reward port; and a psychometrics testing stage (S4), in which mice were tested with six different stimuli, including the two extremes presented in S3. *Figure 1C* illustrates the learning performance for one mouse, showing the time to reach 70% correct trials and the performance at 21 days, as estimated from a linear fit to the daily average performance. Performance starts around 50% correct (chance level for the binary choice) and reaches a level above 80% by the end of stage S3. During this training stage, animals were limited to 500 trials in each session to facilitate comparisons

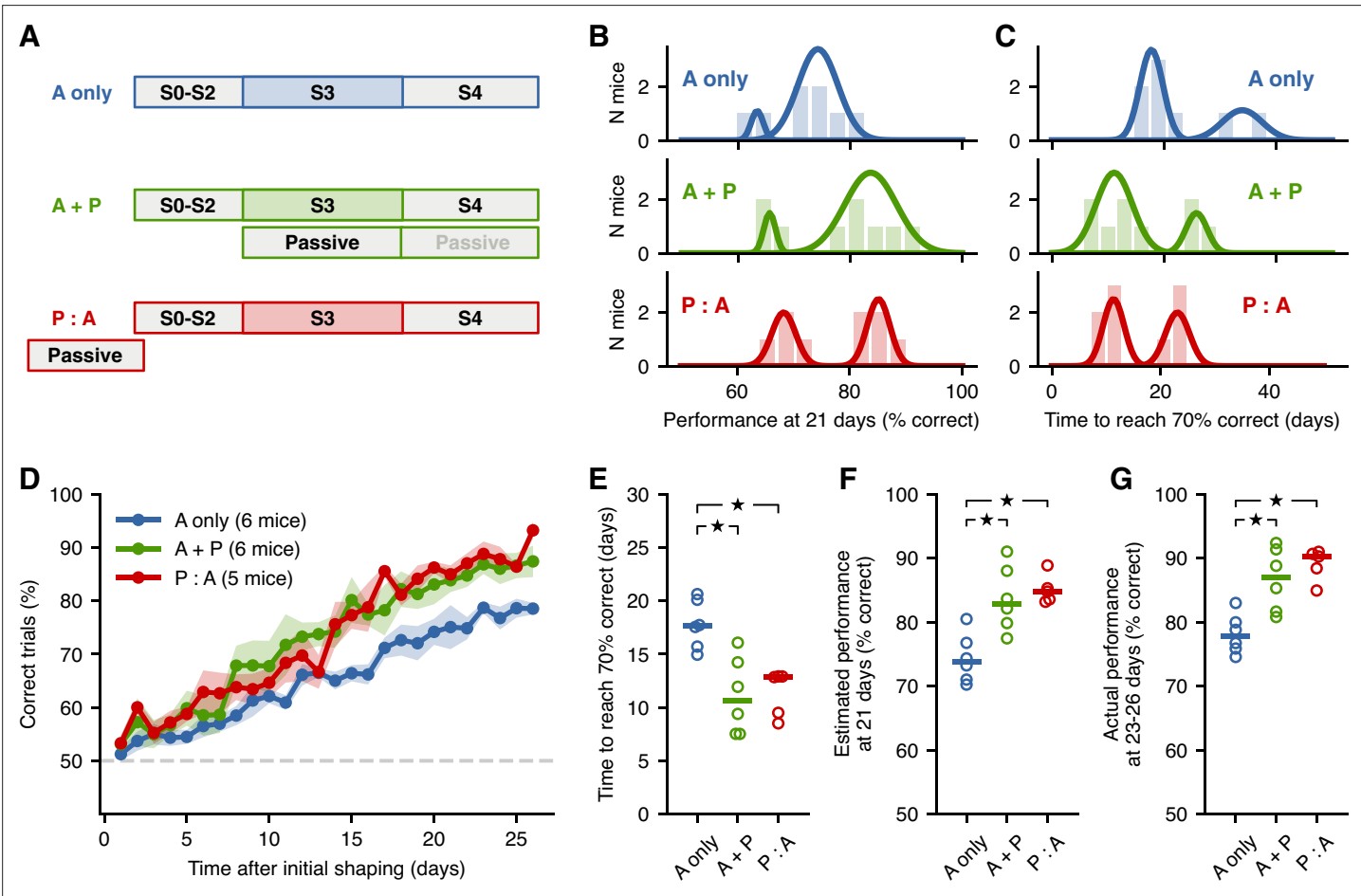

**Figure 2.** Passive exposure to sounds improves learning speed. (**A**) Training schedule for each mouse cohort: A-only mice received no passive exposure; A + P mice received passiveexposure sessions during stage S3; P:A mice received a similar number of passive-exposure sessions before S2. (**B**) Distributions of performance at 21 days of S3 given estimates from linear fits to the learning curve for each mouse from each cohort. Solid lines represent the results from a Gaussian mixture model with two components, separating 'fast' from 'slow' learners. (**C**) Distributions of times to reach 70% correct given estimates from linear fits. (**D**) Average learning curves across fast learners from each cohort. Shading represent the standard error of the mean across mice. (**E**) Estimates of the time to reach 70% for fast learners from each group. Each circle is one mouse. Horizontal bar represents the median. (**F**) Estimates of the performance at 21 days for fast learners from each group. (**G**) Actual performance averaged across the last 4 days of S3 for fast learners from each group. Stars indicate $p < 0.05$.

**Table 1.** Hyperparameters used for shown results.

| | Model 1 | Model 2 | Model 3 | Model 4 | Model 5 |
|---|---|---|---|---|---|
| $\eta_{v,UL}$ | $1 \cdot 10^{-4}$ | $5 \cdot 10^{-2}$ | – | – | $1 \cdot 10^{-5}$ |
| $\lambda_{v,UL}$ | 2.4 | 1 | – | – | 1 |
| $\eta_{v,SL}$ | $2 \cdot 10^{-4}$ | – | $1 \cdot 10^{-2}$ | $3 \cdot 10^{-3}$ | $2 \cdot 10^{-3}$ |
| $\lambda_{v,SL}$ | $5 \cdot 10^{-2}$ | – | $2 \cdot 10^{-2}$ | $3 \cdot 10^{-2}$ | $5 \cdot 10^{-2}$ |
| $\eta_{W,UL}$ | – | – | $2 \cdot 10^{-5}$ | $8 \cdot 10^{-6}$ | $6 \cdot 10^{-5}$ |
| $\lambda_{W,UL}$ | – | – | 1 | 1 | 4 |
| $\eta_{W,SL}$ | – | $1 \cdot 10^{-3}$ | – | – | $1 \cdot 10^{-4}$ |
| $\lambda_{W,SL}$ | – | $1.10^{-3}$ | – | – | $1 \cdot 10^{-3}$ |

across animals. Overall, the number of trials in which animals made no choice after initiating a trial was negligible (averaging 0.18% of trials across mice). *Figure 1D* illustrates the performance of the same mouse during one session of the psychometrics testing stage (S4). As expected, stimuli with FM slopes closer to zero result in responses closer to chance level (50%).

## Passive exposure to sounds improves learning

To test whether passive exposure to sounds enhances acoustic categorization learning of these sounds, we created three cohorts of mice (*Figure 2A*). The first cohort, named 'active training only' (A only), followed the training schedule described above with no additional exposure to the sounds. The second cohort, 'active training with passive exposure' (A + P), received additional passive exposure to sounds during stages S3 and S4. The last cohort, 'passive exposure before active training' (P:A), received passive exposure to sounds before starting the main learning stage S3. Passive exposure for the A + P and P:A cohorts consisted of additional presentation of all six sounds used in S4, randomly ordered, while animals were in their home cages inside a sound isolation booth. Animals received an average of about 3600 passive trials each day, corresponding to 600 daily passive presentations of each of the six stimuli. The amount of passive exposure for P:A mice matched what A + P mice received during stage 3. During stage S4, A + P mice received additional passive sessions.

We evaluated the learning performance of animals from each cohort by fitting a straight line to the daily performance of each mouse during S3 and estimating two quantities from these fits: the performance at 21 days (*Figure 2B*), and the number of days required to reach 70% of trials correct (*Figure 2C*). The distributions of these estimates suggested that each cohort included two types of learners: one group of fast learners and one group of slow learners. To test whether this bimodality was indeed present, we applied a Gaussian mixture model to the data in *Figure 2B* and compared the Bayesian information criterion (BIC) for models with $k = 1$ vs. $k = 2$ Gaussian components. When comparing BIC, models with lower BIC are generally preferred. We found that, for both cohorts that included passive exposure, the BIC for $k = 2$ was lower than for $k = 1$ (*Table 1*), indicating that a model with two components best captured the data. For the cohort with no passive exposure, the BIC was lower when using a single component. However, to make comparisons across cohorts more equitable, we applied the two-component Gaussian mixture model to all cohorts and focused further analysis on the group of faster learners from each cohort. The mice that were categorized as fast learners based on their performance at 21 days (*Figure 2B*) were the same mice that were categorized as fast learners based on the number of days to reach 70% performance (*Figure 2C*).

As a first test of whether passive exposure influenced learning speed, we quantified the average learning curve across fast learners from each cohort (*Figure 2D*). The learning curves show a clear improvement for animals that had passive exposure compared to those who did not. These curves, however, suggested no differences between animals that had interleaved passive-exposure sessions during active-training days (A + P) and animals that had all of their passive exposure occur before the main learning stage (P:A). A quantification of these effects across animals confirmed these observations. Specifically, the time to reach 70% correct trials (*Figure 2E*) was shorter for animals that

experienced passive exposure to sounds compared to those who did not ($p = 0.01$ for A only vs. A + P, $p = 0.006$ for A only vs. P:A, Wilcoxon rank-sum test), while we found no statistically significant difference between the A + P and P:A cohorts ($p = 0.71$, Wilcoxon rank-sum test). A similar result was observed for the estimated performance at 21 days (*Figure 2F*), where animals with passive exposure showed better performance ($p = 0.01$ for A only vs. A + P, $p = 0.006$ for A only vs. P:A, Wilcoxon rank-sum test), while we found no difference between the A + P and P:A cohorts ($p = 0.47$, Wilcoxon rank-sum test). These observations point to an unexpected result: the performance of animals after a few interleaved passive-exposure sessions was as high as that for animals that had received all passive exposures before learning the task, suggesting an interaction between passive exposure and active learning.

To ensure that the apparent effects of passive exposure were not the result of using a simple linear fit to the learning data, we also compared the performance of each animal averaged over the last 4 days of the learning stage (*Figure 2G*). Comparisons across cohorts matched those observed from the linear fit estimates, where mice with passive exposure displayed higher performance ($p = 0.01$ for A only vs. A + P, $p = 0.006$ for A only vs. P:A, $p = 0.71$ for A + P vs. P:A, Wilcoxon rank-sum test). An analysis of the slow learners from each cohort revealed similar trends, where the performance on the last 4 days for animals with only active sessions was lower (66.5% across two mice) than for mice with passive exposure (70.4% across three A + P mice and 76.5% across four P:A mice), although these differences were not statistically significant (p-values in the range 0.064–0.16, Wilcoxon rank-sum test). Overall, these results indicate that passive exposure to task-relevant sounds—either before or during learning—can enhance acoustic categorization learning in adult mice, and they point to a

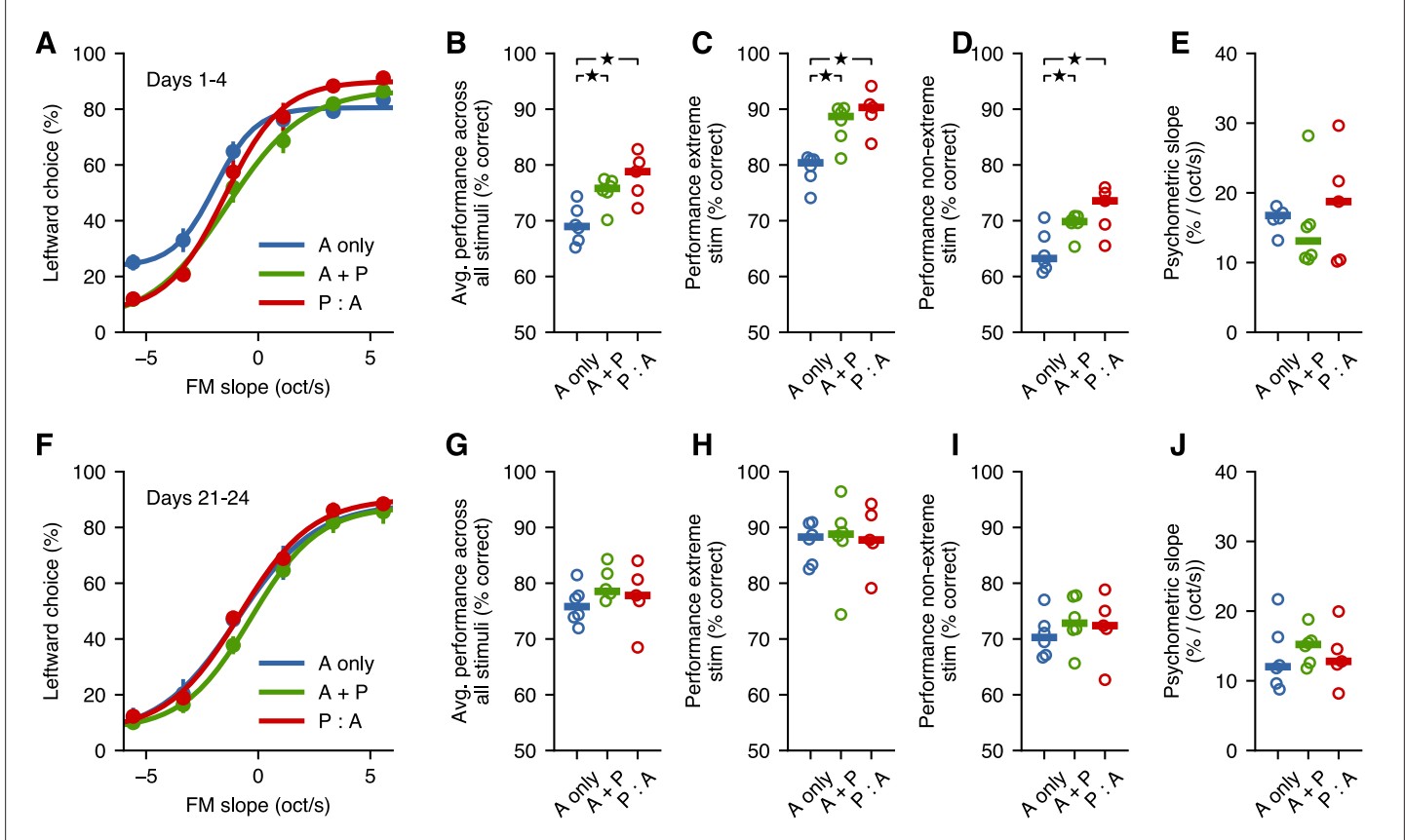

**Figure 3.** Passive exposure improves categorization of intermediate stimuli. (**A**) Average psychometric performance for the first 4 days of stage S4 across fast learners from each group. Error bars show the standard error of the mean across mice. (**B**) Performance averaged across all stimuli is better for mice with passive exposure. Horizontal lines indicate median across mice. (**C**) Performance for extreme stimuli (included in S3) is better for mice with passive exposure. (**D**) Performance for intermediate stimuli (which were not used in the task before S4) is better for mice with passive exposure. (**E**) Psychometric slope is not different across groups. (**F–J**) All groups achieve similar levels of performance after 3 weeks of additional training. Stars indicate $p < 0.05$.

non-trivial interaction between passive exposure and active learning, which we further investigate in our theoretical analysis below.

## Passive exposure influences responses to intermediate sounds not used during training

To test whether passive exposure influenced the behavioral responses to sounds beyond those used during the active-training sessions, we evaluated the psychometric performance of animals from each cohort during stage S4. We first estimated the average psychometric curves across all mice from each cohort during the first 4 days of S4 (*Figure 3A*). These curves illustrate that, as expected from the results of S3, the performance on the extreme sounds is better for animals that received passive exposure. Moreover, these curves hinted at differences across cohorts in the responses to intermediate sounds, which were presented during passive exposure, but had not been part of the active training during the learning stage. To quantify these effects, we first measured the average performance across all stimuli and found that animals that experienced passive exposure achieved a higher fraction of correct trials overall compared to those that did not experience passive exposure ($p = 0.01$ for A only vs. A + P, $p = 0.01$ for A only vs. P:A, $p = 0.27$ for A + P vs. P:A, Wilcoxon rank-sum test) (*Figure 3B*). This disparity in overall performance was the result of differences in both the responses to the extreme sounds (*Figure 3C*) as well the responses to intermediate sounds ($p = 0.037$ for A only vs. A + P, $p = 0.028$ for A only vs. P:A, $p = 0.36$ for A + P vs. P:A, Wilcoxon rank-sum test) (*Figure 3D*). To evaluate the possibility that having no passive exposure resulted in shallower psychometric curves, we compared the psychometric slopes across cohorts. We found no significant difference in psychometric slopes across cohorts ($p = 0.07$ for A only vs. A + P, $p = 1$ for A only vs. P:A, $p = 0.86$ for A + P vs. P:A, Wilcoxon rank-sum test, *Figure 3E*). This observation, together with the changes in performance for extreme stimuli, suggests that the observed effects of passive exposure are not simply captured by psychometric curves becoming less shallow. Overall, these observations indicate that passive exposure can have an effect on the behavioral responses to stimuli beyond those used during the active training sessions.

To test whether the asymptotic performance of animals was affected by passive exposure, we compared the psychometric performance across cohorts after 21 days of S4 sessions. We found that animals with no passive exposure improved their average performance during these few weeks ($p = 0.031$ when comparing early and late periods of S4, Wilcoxon signed-rank test), while other cohorts did not change in any consistent manner, suggesting that most of these mice had already reached asymptotic performance ($p = 0.56$ for A + P, $p = 1.0$ for P:A, Wilcoxon signed-rank test). During this period, animals with no passive exposure improved until they were indistinguishable from those with passive exposure ($p > 0.26$ for all comparisons, Wilcoxon rank-sum test) (*Figure 3F–J*). These results indicate that the differences observed across cohorts were not the result of specific sets of animals having predisposition for poorer learning, but rather, that passive exposure speeds up learning performance without an apparent change in final performance.

## A one-layer model does not benefit from passive pre-exposure

In the experiments described above, we found that passive exposure to task-relevant stimuli benefits learning, regardless of whether it occurred before or in-between active-training sessions. In order to gain insight into the neural mechanisms that might underlie this observation, we analyzed the effects of active learning and passive exposure in a family of artificial neural-network models combining supervised and unsupervised learning. Specifically, we evaluated the consequences of different learning algorithms, learning schedules, network architectures, and stimulus distributions on the learning outcomes. To simulate frequency-modulated sound inputs, we provided our models with input representations $\vec{x} = \vec{\mu} + \vec{\xi}$, where $\vec{\mu}$ deterministically encodes the stimulus, and $\vec{\xi}$ is isotropic noise with covariance $\sigma_x^2 \mathbf{I}$. The binary output of the models corresponds to the choice to lick the left or right port in the experiment. We assumed that one direction in the space of all possible input representations corresponds to the FM slope parameter. The sounds with the most extreme FM slopes presented to the models lie at the points $\pm\vec{\mu}_0$, and, as for the mice in the experiment, the task the models were tested on was to associate these two most extreme input representations with the labels $\pm 1$. We trained the models with combinations of:

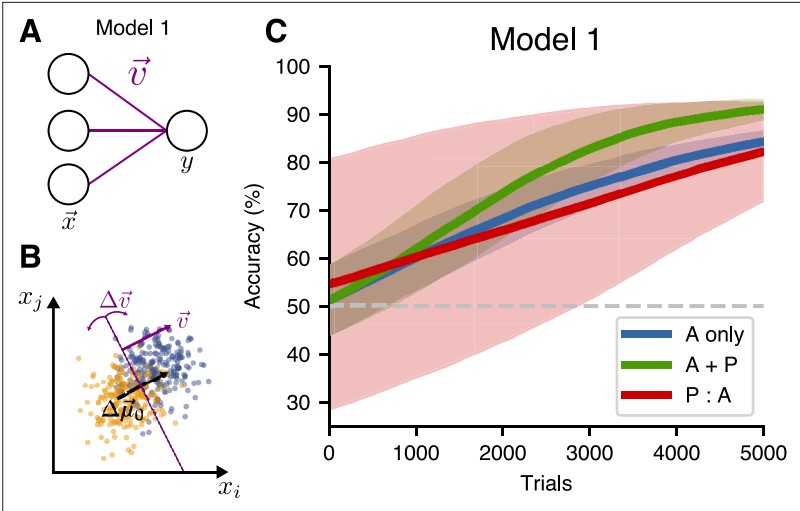

**Figure 4.** A single-layer model (Model 1) does not benefit from passive pre-exposure. (**A**) Network architecture for the one-layer model. (**B**) The network is trained to find a hyperplane orthogonal to the decoding direction $\Delta\vec{\mu}_0$. (**C**) Learning performance for different training schedules. Curves show mean accuracy for $n = 50$ network realizations, and shading shows standard deviation.

- Active learning: The model was provided with a sample from the extremes of the stimulus distribution and the corresponding sample label.
- Passive exposure: The model was provided with a sample but no label. To replicate the passive-exposure sessions in the experiment, these passive samples were drawn from normal distributions with means at six points on the line segment between $+\vec{\mu}_0$ and $-\vec{\mu}_0$ and noise covariance $\Sigma$.

We aimed to find models that can replicate the general experimental observation that passive exposure improves learning speed. Thus, all the models we considered included some parameters that were trained using unsupervised learning, which does not require feedback about task performance, during the active and passive training sessions. Since the models also had to learn the association between labels and stimuli, they also needed to include parameters that were trained during active sessions using supervised learning, which makes use of feedback about task performance.

Active trials and passive exposures were combined into the following three training schedules:

- Active only (A only): The model was always provided with a sample and its label during 5000 total trials of training.
- Active and passive (A + P): The model also underwent 5000 active trials, but each one was followed by 9 passive exposures.
- Passive then active (P:A): The model was first presented with 45,000 passive exposures, then underwent 5000 active trials.

The first model (Model 1) we considered was the simplest possible neural-network model, which consisted of a single output neuron reading out from an input representation (**Figure 4A**). In this one-layer model, the input representation $\vec{x}$ was multiplied by a weight vector $\vec{v}$ to produce an output $S(\vec{v} \cdot \vec{x})$, where $S$ is the logistic sigmoid function. The task the model had to perform is illustrated in **Figure 4B**: the weight vector $\vec{v}$ is orthogonal to the decision hyperplane, so its optimal orientation would be along the direction $\Delta\vec{\mu}_0 = +\vec{\mu}_0 - (-\vec{\mu}_0)$. Accounting for the experimental data requires a model that changes even when no labels are provided during passive exposure. Therefore, we trained the model with both *supervised learning* (in which the weight $\vec{v}$ undergoes gradient descent, equivalent to logistic regression) and *unsupervised learning* (where the weight $\vec{v}$ is trained with Hebbian learning with weight decay using Oja's rule *Oja, 1982*). The unsupervised learning rule used here aligns the normal vector to the decision hyperplane with the direction of highest variance in the input representation, which in this case is the coding direction spanned by $\Delta\vec{\mu}_0$. For training, we used the three learning schedules introduced above. During the active sessions, the weight $\vec{v}$ was trained by

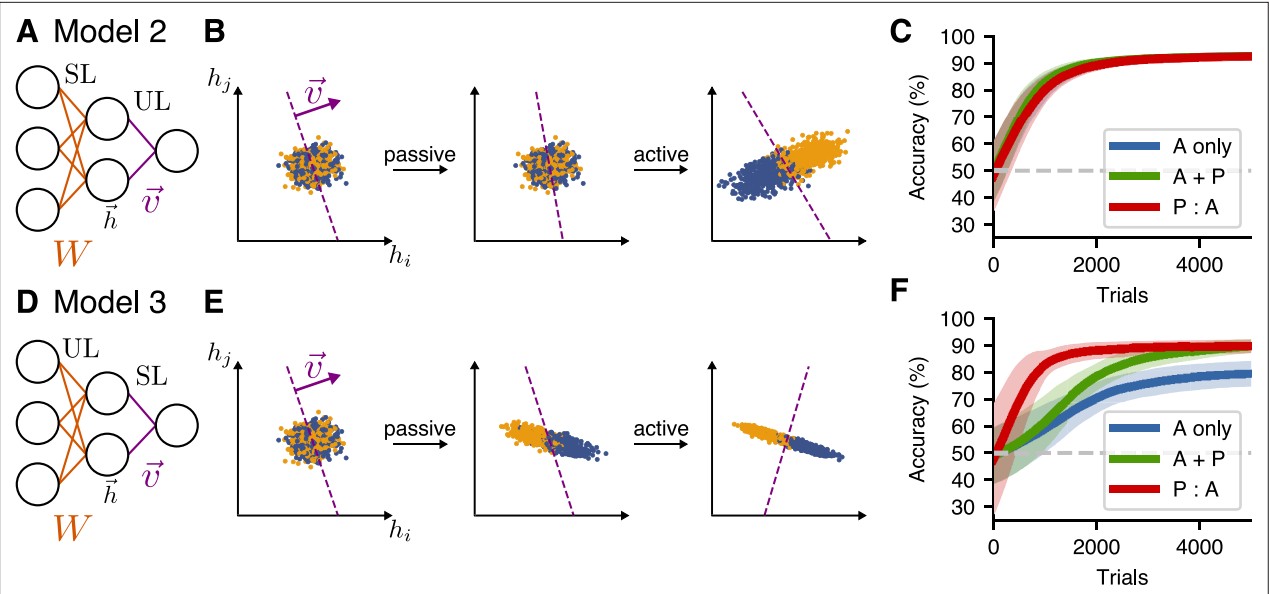

**Figure 5.** Passive-exposure benefits learning in a two-layer model with unsupervised learning in the first layer. (**A**) Network architecture for Model 2, which has supervised learning (SL) at the input layer, and unsupervised learning (UL) at the readout. (**B**) Learning dynamics of the hidden-layer representation of this model for the P:A schedule. (**C**) Learning performance for different training schedules for Model 2. Curves show mean accuracy for $n = 50$ network realizations, and shading shows standard deviation. (**D–F**) Model 3, which has unsupervised learning (UL) at the input layer, and supervised learning (SL) at the readout.

both supervised and unsupervised learning, while, during the passive sessions, only the unsupervised learning rule was used.

As in the experiment, interleaving active trials with passive exposure (the A + P condition) facilitated learning and slightly sped up training relative to active-only training (*Figure 4C*). Long passive pre-exposure before active learning (P:A condition), however, did not contribute to task learning in this model. This can be explained by the symmetry of the task: the decision hyperplane oriented itself in the optimal direction to separate the point clouds, however, because the algorithm did not know about the data labels, there was a 50% chance it was correctly oriented, and a 50% chance it was oriented exactly in the wrong direction. These two possible configurations of the model after the passive pre-exposure sessions averaged out, giving no net benefit to the P:A schedule over the active-only training. Because of this failure, the one-layer model cannot capture the experimental observation that passive pre-exposure improves the speed of learning.

## Passive exposure is beneficial when building latent representations with unsupervised learning

To remedy this shortcoming, we studied a simple extension to the above model by adding an additional layer of hidden neurons. In addition to the readout weights $\vec{v}$, this two-layer model had initial weights $W$ mapping the input representation $\vec{x}$ to a hidden representation $\vec{h} = W\vec{x}$. The output of the model was then $S(\vec{v} \cdot W\vec{h})$. For our simulations, the dimension of the hidden layer $d_{\text{hid}}$ was smaller than the input dimension $d$. We trained $\vec{v}$ using the same algorithms as in the one-layer model. In addition, we have the option to train $W$ with either supervised learning, unsupervised learning, or both. As for the one-layer model, we needed to include both supervised and unsupervised learning to account for the experimental observations. The simplest way to incorporate this is to have supervised learning in one layer and unsupervised learning in the other one.

First, we investigated what happens if we use supervised learning for the input weights $W$ and unsupervised learning for the readout weights $\vec{v}$. This defines Model 2 (*Figure 5A*). In this model, the relative learning performance for all schedules was similar to that of Model 1 (*Figure 5C*). This can be understood by observing that, since the hidden-layer representation did not change during passive exposure in the P:A schedule, the active sessions started from a representation that did not allow the

signal to be decoded, as illustrated in *Figure 5B*. Thus, there was no benefit from the initial passive sessions in this model, which contradicts the experimental outcomes. Here, we do not make claims about the relative performance of the learners in Model 2 with those of Model 1 (since the hyperparameters of each model were chosen independently such that they give rise to similar asymptotic accuracies). Instead, the conclusions are drawn by comparing different learning schedules for a given model.

Another possibility (Model 3) to incorporate both types of learning in this network is to learn the input weights $W$ by unsupervised learning and the readout weights $\vec{v}$ using supervised learning (*Figure 5D*). Doing so allowed the passive exposures to build a representation of the first principal component of the input representation, which enhanced the decodability in the hidden layer (*Figure 5E*). For this model, the speed of learning for both the P:A and A + P training schedules increased compared to the active-only case (*Figure 5F*), accounting for the main feature of behavioral experiments, namely, that passive exposure enhances learning speeds.

While this general behavior is consistent with the experimental results, two main issues remain to better account for the data. First, the learning curve for the P:A learners in *Figure 5D* rises very quickly compared to the A + P models (which built up the same representation more gradually during the active trial period). In contrast, these two conditions had similar learning speeds in our experiments. Second, because of the unsupervised learning rule chosen for Models 1–3, the coding direction of the neural representation of stimuli must align with the first principal component of the neural representation for the system to benefit from passive exposure. In general, however, features that are relevant for learning will not always be encoded along the first principal component. The following section presents solutions to both of these issues.

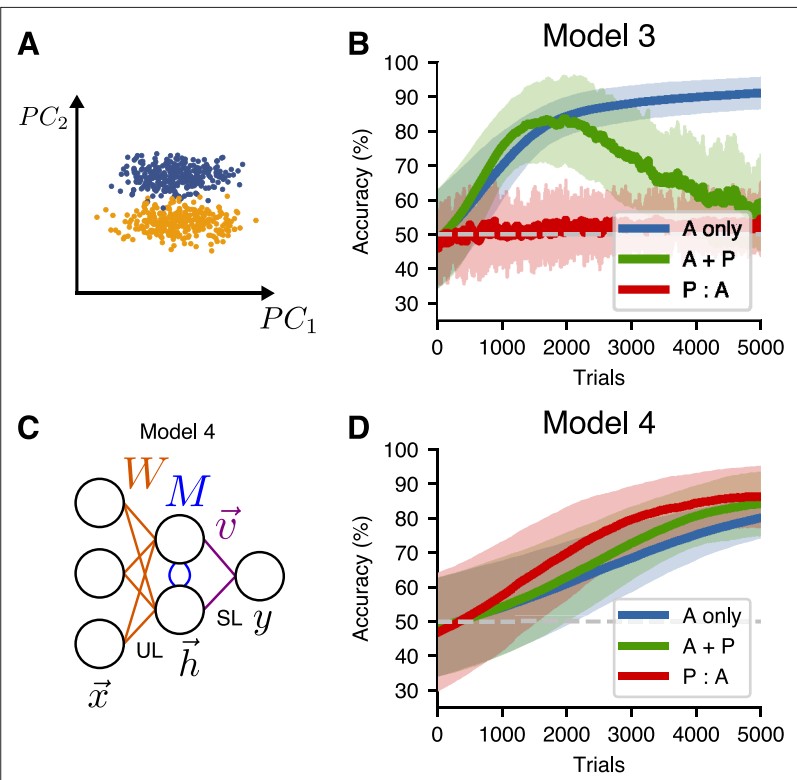

**Figure 6.** An alternative unsupervised learning rule can be used to learn hidden representations of higher principal components. (**A**) A non-isotropic input distribution in which the coding dimension does not align with the direction of highest variance. (**B**) Learning performance for Model 3 on the non-isotropic input distribution. Curves show mean accuracy for $n = 50$ network realizations, and shading shows standard deviation. (**C**) Network architecture for a two-layer model (Model 4) that uses the similarity matching algorithm for the input weights and supervised learning at the readout. (**D**) Learning performance for Model 4 on the non-isotropic input distributions.

## Building higher-dimensional latent representations improves learning for more-general input distributions

A limitation of Model 3 is that Hebbian learning only builds a representation of the direction of highest variance (the first principal component) of the input representation. In real neural representations, however, the coding direction will not always align with the first principal component. To investigate what happens in this case, we trained Model 3 on a non-isotropic input distribution, setting $\Sigma = \mathrm{diag}(\sigma_1^2, \sigma_2^2, \sigma_2^2, \ldots)$ with $\sigma_1 \gg \sigma_2$. We set the extreme means $\pm\vec{\mu}_0 = \pm\vec{e}_2$, where $\vec{e}_i$ is the unit vector along the $i$th dimension. For these inputs, the coding direction represents the second principal component (*Figure 6A*). In this case, passive exposure is counterproductive for learning because the variance in the coding direction is lost in the hidden-layer representation, so it cannot be used to distinguish the two input distributions, leading to worse performance of A + P and P:A learners relative to A-only learners (*Figure 6B*).

To address this issue, we implemented a model that uses the *similarity matching* unsupervised learning algorithm (*Pehlevan et al., 2015*) to build a higher-dimensional hidden representation that includes higher principal components. To do this, we modified the network to include lateral weights $M$ connecting the hidden units in our two-layer model (*Figure 6C*). These new connections were trained using anti-Hebbian learning and have the function of decorrelating the hidden units.

With these modifications, which define Model 4, the performance (*Figure 6D*) was similar to that of Model 3 with the isotropic input distribution (*Figure 5F*). Specifically, in the P:A schedule, the representation built up during passive sessions aided decodability, slightly increased the speed of learning, and led to a higher final performance relative to the A-only schedule. However, while the P:A learning performance exhibited some improvement over the A-only performance, the A + P learning performance did not show a comparable improvement for any set of hyperparameters that we investigated (see Methods). This is because, in the P:A schedule, the active training benefits from a representation that aids decoding, while this representation must be built up over time for the learners in the A + P schedule. Thus, the P:A curve in *Figure 6D* initially rises faster than the A + P curve, unlike in the experimental outcomes shown in *Figure 2D*. Together, these results show that, in the case where the task-relevant encoding direction is not aligned with the direction of highest variance in the input representation, a more-sophisticated unsupervised learning algorithm is capable of accounting for the enhanced learning due to passive exposure, but not for the similarity in the improvements exhibited by the A + P and P:A training schedules.

So far, we have only considered input stimulus representations for which the decoding direction lay in the subspace spanned by the highest principal components, such that unsupervised learning at the input layer is sufficient to create an optimal latent representation. In a more natural setting, parts of the decoding direction might be aligned with the highest-variance principal components, but part of it might not be. To study this case, we created a model (Model 5) with two new features. First, this model receives an input representation in which the coding direction has a nonzero projection along the first 30 principal components. Second, because an unsupervised learning rule that finds leading principal components alone is insufficient to reach an optimal solution when the signal is not entirely contained within the leading principal components, this model combines both supervised and unsupervised learning at the input layer (*Figure 7A, B*). Compared to Models 3 and 4, P:A and A + P schedules led to similar improvements over A-only training (*Figure 7C*), consistent with the experimental results shown in *Figure 2D*.

In addition, when we tested the performance stimulus values for all six stimuli at different points in learning, this model reproduced the main features of the psychometric curves from the experimental data. In particular, midway through training we found that the psychometric curves for the A + P and P:A learners had fully converged and were mostly overlapping, while the A-only learners exhibited relatively poorer classification performance for all stimulus values (*Figure 7D*; *Figure 3A*). At the end of training, all three curves converged to very similar values for all stimulus values (*Figure 7E*; *Figure 3F*). Thus, Model 5 reproduced all of the key findings from the experiments, including the behaviors of the learning curves and psychometric curves in all three training conditions.

In Models 3 and 4, we found that P:A learning was initially faster than A + P learning, which we attributed to the larger number of representation-improving passive exposures that the P:A learners received in the early phases of training. Somewhat surprisingly, we then found in Model 5 that the P:A and A + P learning curves rose comparably quickly during the initial phase of active training. We

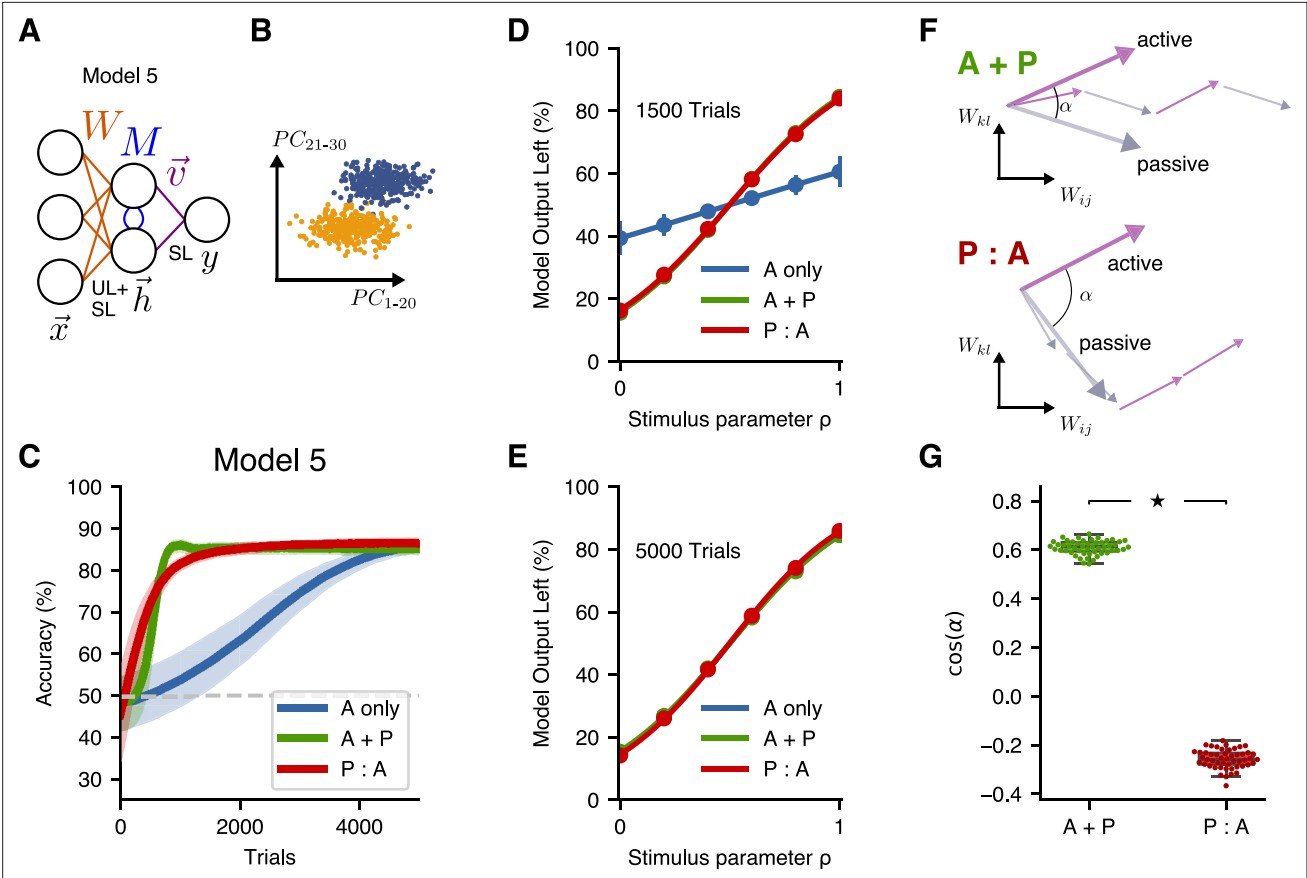

**Figure 7.** A more-general two-layer model accounts for similar benefits of A + P and P:A training schedules. (**A**) Schematic illustration of Model 5, which combines supervised and unsupervised learning at the input layer of weights. (**B**) Input distribution, in which the decoding direction is not aligned with any particular principal component. (**C**) Learning performance of Model 5. Curves show mean accuracy for $n = 50$ network realizations, and shading shows standard deviation. (**D**) Psychometric curves showing classification performance for all stimuli after 1500 trials, where the stimulus parameter $\rho$ linearly interpolates between the two extreme stimulus values. (**E**) Psychometric curves showing classification performance after 5000 trials. (**F**) Schematic illustration of the angle between the summed weight updates during active training and passive exposure for A + P (top) and P:A (bottom) learners. (**G**) The alignment of active and passive weight updates for $n = 50$ networks trained with either the A + P or P:A schedule after 1500 trials (star indicates $p < 10^{-3}$, Wilcoxon rank-sum test; box percentiles are 25/75).

hypothesized that this occurred due to an improved alignment in the weight updates during active learning and passive exposure in the A + P case (in which these updates occur in alternation) vs. the P:A case (in which all of the passive-exposure updates occur before the active-training updates). To test this, we computed the angle of the sum of all active updates relative to the sum of all passive updates in A + P and P:A learners (*Figure 7F*). Consistent with our hypothesis, we found that the active and passive updates were more aligned for the A + P learners than for the P:A learners (*Figure 7G*). This result establishes a potential mechanism by which an interleaved schedule of active training and passive exposure leads to more-efficient learning (in the sense of requiring, for a given number of active-training steps, fewer passive exposures to achieve a given performance) than a schedule in which passive exposure entirely precedes active training.

Together, our experimental and theoretical results have shown that the experimentally observed benefit of passive exposure in both the P:A and the A + P schedules is consistent with neural-network models that build latent representations of features that are determined by statistical properties of the input distribution, as long as those features aid the decoding of task-relevant variables.

## Discussion

In this work, we have shown that passive exposure to task-relevant stimuli increases the speed of learning during active training in adult mice performing a sound-categorization task. Specifically, for the amount of passive exposure used here, we found similar increases in cases where the passive exposure occurred before active training or interleaved with active training, even at early points where the cumulative number of passive exposures in the latter case was far smaller. Using artificial neural networks, we showed that these results are consistent with a multi-layer model in which unsupervised learning in an early layer creates a latent hidden representation that reflects the statistics of the input stimuli, and supervised learning in a later layer is then used to decode the stimulus properties and map them onto appropriate behaviors. Finally, we found that improved learning efficiency when passive exposure is interleaved with active training rather than occurring entirely before active training can be accounted for by active and passive weight updates adding together more constructively in interleaved training—a result that may have implications for designing optimal training schedules in humans, animals, and artificial neural networks.

Various lines of research have investigated the idea that exposure to stimuli may influence perceptual judgments. Multiple studies have demonstrated that the statistics of sensory stimulation during an animal's development have a strong influence on the perceptual abilities (and associated neural correlates) in the adult (*Hensch, 2004*). Other studies in adults have focused on perceptual learning, generally defined as experience-dependent enhancements of the ability to perceive and discriminate sensory stimuli during perceptual decision tasks (*Gold and Watanabe, 2010*). These studies have shown decreases in the strength, quality or duration of a stimulus needed to obtain a particular level of accuracy, as animals get more exposure to the task stimuli. A key observation from these studies is that these effects can be disambiguated from other forms of learning, such as those that establish task rules. In addition to these perceptual enhancements, learning related to specific stimulus features can occur even when subjects are not told of the relevance of these features for a given task. Studies of this phenomenon, usually called 'incidental learning', have shown for example that subjects can incidentally learn categories of complex acoustic exemplars that occur before visual stimuli, even when the instructed task is visual detection (*Gabay et al., 2015*). Beyond these effects of incidental learning, studies in humans have found that, under specific conditions, passive exposure to sounds interleaved with training can be beneficial for learning, sometimes to the extent that active sessions can be replaced with passive exposure and still yield similar performance (*Wright et al., 2015*)—an effect that was later replicated for olfactory learning in mice (*Fleming et al., 2019*). Our experimental results complement these observations by demonstrating that passive exposure in adult mice (either interleaved with training or before training) enhances the learning of acoustic categories, opening new avenues for the detailed investigation of the neural mechanisms of the improvements that result from passive exposure in audition.

One important question to address in future studies is to which extent the passive stimuli need to be related to those presented in the task. Related to this point, previous work has shown that sensory enrichment alone can change cortical sensory maps (*Polley et al., 2004*) and improve task performance (*Mandairon et al., 2006*; *Alwis and Rajan, 2014*). Beyond the effects on learning and perceptual judgment, studies in rodents have shown that the effect of stimulus pre-exposure in classical conditioning paradigms can vary depending on test procedures, the similarity of pre-exposure and training procedures, and the choice of response measure (*de Hoz and Nelken, 2014*; *Holland, 2018*). This suggests that learning associations related to reward or punishment and the perceptual enhancements that come from passive exposure may rely on different mechanisms and, under some conditions, compete with each other (*McLaren et al., 1989*). Therefore, the design of schedules that benefit optimally from passive exposure must take these mechanisms into account. Given these constraints, a better understanding of the neural mechanisms underlying the influence of passive exposure, achievable through a combination of theoretical approaches and experiments in animals that provide sufficient experimental access, have the potential to guide the design of appropriate schedules in a more efficient manner compared to behavioral experiments alone.

In our experiments, we found that providing animals with passive exposure before task training vs. interleaved with task training led to comparable benefits. This unexpected result could be a consequence of the large number of passive-exposure trials provided to the animal on each day. A comprehensive evaluation of the effects on learning performance as a function of number of passive

exposures may be needed to test this hypothesis. Our models, in contrast, most often found that exposure before task training led to larger gains (Models 3 and 4), compared to interleaved exposure, although we also found a model that led to comparably large gains in these two cases (Model 5). These theoretical results suggest that different schedules of passive exposure and active training might lead to significant differences in learning performance, and future experimental work could test whether this in fact occurs.

Our models make the experimental prediction that stimulus features should become more easily decodable from neural representations following frequent exposure to those stimulus features, even before those stimuli have occurred within the context of a learned task. Related to this idea, previous work has shown that neural responses in primary sensory cortices exhibit within-session adaptation to stimulus statistics (*Dean et al., 2005*; *Sharpee et al., 2006*; *Garcia-Lazaro et al., 2007*; *Gutnisky and Dragoi, 2008*). However, less is known about how such within-session adaptation relates to long-term plasticity occurring across days. Future experiments that include recording throughout task learning could test whether neural representations evolve in a manner consistent with our models.

In the machine-learning literature, various approaches to combine labeled and unlabeled data in a semi-supervised learning classification algorithm have been put forward (*van Engelen and Hoos, 2020*), including some biologically plausible implementations (*Gu et al., 2019*; *Genkin et al., 2019*). In most of these, unlabeled data are either used for regularization (e.g., *Belkin et al., 2005*) or are assigned pseudo-labels and then used for training (*Triguero et al., 2015*). In contrast, in our model we used unsupervised learning in an early layer to create a useful representation for supervised learning downstream, a simple form of semi-supervised feature learning.

One limitation of our modeling approach is that the set of models that we consider does not include some features that may be important for fully capturing the mechanisms of unsupervised learning during passive exposure in the brain. More-sophisticated approaches beyond learning rules that implement linear dimensionality reduction will be required for cases in which relevant stimulus features are encoded in highly nonlinear ways, as would likely be the case for natural sound stimuli. If the input statistics are very complex, simple forms of initial unsupervised learning such as the ones that we used might not be helpful for improving hidden-layer representations and learning (*Iyer et al., 2020*). Recent years have seen tremendous advances in addressing this challenge by the use of self-supervised learning to learn complex stimulus features in deep neural networks (e.g., *Devlin et al., 2018*; *Avd et al., 2018*; *Grill et al., 2020*). While the models that we have presented make simplifying assumptions about the stimulus statistics and learning rules, we conjecture that the principle they are meant to illustrate—namely, that unsupervised learning can make subsequent supervised learning more efficient by improving neural representations—applies broadly across different stimulus statistics and learning rules.

## Methods

### Animal subjects

A total of 27 wild-type C57BL/6J adult mice (RRID:IMSR_JAX:000664) of both sexes, ages 2.5–4 months at the beginning of behavioral training, were used in this study. All mice were housed in groups of same-sex littermates in a 12:12-hr light–dark cycle. Experiments were carried out during the dark period, when mice are most active. Mice were water restricted to motivate them to perform the behavioral task. Mice were weighed and their health checked after each behavioral session, and they were provided with a water supplement if their weight was below 80% of their baseline. Except for these supplements, access to water was restricted to the time of the task during experimental days. Free water was provided on days with no experimental sessions. All procedures were carried out in accordance with National Institutes of Health Standards and were approved by the University of Oregon Institutional Animal Care and Use Committee (Protocol #21-26).

The behavioral data were collected using the taskontrol software platform (https://taskontrol. readthedocs.io) written in the Python programming language (https://www.python.org/). Freely moving mice were trained to discriminate whether the slope of a 200-ms frequency-modulated sound was positive or negative. Animals initiated each trial by poking the center port of a three-port chamber, at which point the sound was presented after a brief silent delay (150–250 ms, uniformly distributed). Mice then had to choose the left or right reward port depending on the slope of the stimulus: left

for an upward frequency sweep and right for a downward sweep. Animals were allowed to withdraw before the end of the sound to make a choice, and had up to 4 s after the end of the sound to enter a side port. If a mouse did not respond in this period of time, the trial was aborted and not considered during data analysis. Correct choices were rewarded with a 2-µl water, while incorrect choices yielded nothing and animals had to start a new trial by poking again in the center port.

Animals were first trained with frequency-modulated sounds that spanned the frequency range from 6 to 13 kHz, resulting in an FM slope of ±5.6 oct/s. To evaluate psychometric performance, the frequency range was varied to achieve intermediate FM slopes (3.4 and 1.1 oct/s), while keeping the duration of the sounds and the middle frequency constant. All sounds were presented at an intensity of 70 dB SPL.

## Training stages

Mice were trained to perform the task through a sequence of shaping stages and having a single 1-hr behavior session each day. In stage 0, the goal was to familiarize animals with the reward delivery ports. During this stage, whenever an animal poked in the side port corresponding to that trial (the port was randomized on each trial), water was delivered immediately. Animals stayed in this stage for 2 days. The goal of stage 1 was to teach animals that a trial starts by poking in the center. During this stage, whenever the animal poked in the center port, water was delivered immediately in the corresponding side port for that trial. Animals stayed in this stage for 4 days. The goal of stage 2 was to teach animals to wait for the beginning of the sound and only then make a choice by reaching the correct reward side port. If animals reached the incorrect port, they still had a chance to get a reward by going to the other side port within 4 s of the end of the sound stimulus. During this stage, the delay between the center poke and the stimulus was increased by 10ms every 10 trials, starting at 10ms. Animals stayed in this stage until 70% of the mice achieved 300 rewarded trials in a session (corresponding to 12 days for 'A only' and 'A + P' cohorts, and 9 days for the 'P:A' cohort).

Stage 3 was the main learning stage in which animals only got rewarded if they made the correct choice in their first attempt on each trial. During this stage, we implemented a bias-correction method as follows. If the percentage of correct choices on either side was lower than 20%, the next session was set in a mode where error trials were followed by identical trials, until the mouse made the correct choice. Animals were taken off bias correction when the percentage of correct choices for both sides was above 30%. Bias-correction sessions were not included in the analysis of learning speed. Learning performance during stage 3 was evaluated for 26 days.

After the main learning stage, animals transitioned to stage 4 where we evaluated their psychometric performance by introducing four new sounds of intermediate FM slope, for a total of six sounds per session. Which sound was presented on each trial was randomized according to a uniform distribution. The three sounds with positive FM slope were rewarded on the left port, while those with negative FM slope were rewarded on the right port.

## Passive exposure

Mice were grouped into three cohorts: an 'active training only' (A only) cohort, an 'active training with passive exposure' (A + P) cohort, and a 'passive exposure before active training' (P:A) cohort. Animals that eventually formed the first two cohorts were trained simultaneously in stages 0–2. This group was then split into the 'A only' and 'A + P' by selecting animals to match as closely as possible the average initial performance after shaping between the two cohorts. The 'P:A' mice were trained as a separate cohort. Animals in this cohort had free access to water until their active-training sessions started. One mouse that did not perform enough trials in stage 2 was removed from the study and excluded from further analysis. The total number of animals included in each cohort was therefore: eight A only mice, nine A + P mice, and nine P:A mice.

Passive exposure consisted of the additional presentation of all six sounds used in stage 4, randomly ordered, while animals were in their home cages inside a sound isolation booth. Animals received an average of about 3600 passive trials each day, corresponding to 600 daily passive presentations of each of the six stimuli. Stimuli were presented every 4.5 s. During these sessions, animals showed both periods of activity (running, climbing, etc.) and periods of inactivity. For animals in the A + P cohort, passive-exposure sessions took place the same day as active-training sessions, usually a few hours after training.

## Analysis of behavioral data

To characterize the learning performance of each animal, we calculated the percentage of correct trials for each behavioral session during stage 3 and fit a straight line (without constraints) to these data. Using these linear fits, we determined the performance at 21 days and the number of days required to reach 70% of trials correct for each animal. To test for bimodality of the distributions of these estimates, we used a Mixture Gaussian Model (implemented in the scikit-learn Python package: *Pedregosa et al., 2011*).

The psychometric performance for each mouse was estimated by fitting a sigmoidal curve to the percentage of trials with leftward choices for each stimulus averaged across all sessions of interest (days 1–4 or 21–24 of stage 4). The psychometric slope presented in *Figure 3* was determined from the maximum slope of this sigmoidal fit. The average psychometric performance for each cohort was calculated by first estimating the average performance for each stimulus for each animal, and then averaging across animals. To test differences between cohorts we used the non-parametric Wilcoxon rank-sum test. The behavioral data collected in this study are publicly available on Zenodo (*Schmid et al., 2023*).

## Modeling

The neural-network models described in this article were implemented in JAX (*Bradbury et al., 2018*); their source code is available at https://github.com/cschmidat/behaviour-models (*Schmid, 2023*). The networks were trained on inputs drawn from normal distributions $\mathcal{N}(\vec{\mu}, \Sigma)$, parameterized by the mean $\vec{\mu}$. We assume these means all lie on a line segment, ending at the values $\pm\vec{\mu}_0$. We trained the models in three different settings: With *isotropic* input, *non-isotropic* input, and a *non-aligned* input, in which the decoding direction is only partially aligned with the highest principal components. For the isotropic input, we chose an input dimension $d = 50$, and means along $\vec{\mu}_0 = 1.5\,\vec{e}_1$ , where $\vec{e}_i$ denotes the unit vector along the $i$th direction. Because the model is linear, this choice entails no loss of generality. The covariance matrix was chosen to be $\Sigma = \mathbf{I}$. For the non-isotropic input, we set $d = 50$ and $\vec{\mu}_0 = 1.4\,\vec{e}_2$, and the covariance to $\Sigma = \mathrm{diag}(\sigma_1^2, \sigma_2^2, \sigma_2^2, \ldots)$, with $\sigma_1 = 1$ and $\sigma_2 = \sqrt{8}$. For the non-aligned input, we set $d = 100$,

$$\vec{\mu}_0 = \frac{1.5}{\sqrt{30}} \sum_{i=1}^{30} \vec{e}_i,$$

and

$$\Sigma = \mathrm{diag}(\underbrace{\sigma_1^2, \ldots, \sigma_1^2}_{20}, \underbrace{\sigma_2^2, \ldots, \sigma_2^2}_{80}),$$

where $\sigma_1 = \sqrt{2}$ and $\sigma_2 = 1$. With this choice of parameters, the decoding direction is only partially aligned with the first 20 principal component directions of the input distribution. For all three of these input distributions, the optimal performance for a classifier is about 95%.

The models were trained in discrete steps, corresponding to either:

- One passive exposure, where the models were supplied with a sample drawn from $\mathcal{N}(\vec{\mu}, \Sigma)$, with $\mu$ randomly chosen from six regularly interspaced points on the line segment from $-\vec{\mu}_0$ to $+\vec{\mu}_0$.
- One active trial, where the models were supplied with either (1) a sample drawn from $\mathcal{N}(+\vec{\mu}_0, \Sigma)$ with target output $y = 1$ (corresponding to label +1) or (2) a sample drawn from $\mathcal{N}(-\vec{\mu}_0, \Sigma)$ with target output $y = 0$ (corresponding to label −1).

Active trials and passive exposures were combined into the following three training schedules:

- A only: The model underwent 5000 active trials.
- A + P: Each of 5000 active trials was followed by 9 passive exposures.
- P:A: The model was first presented with 45,000 passive exposures, then underwent 5000 active trials.

With these input representations and learning schedules, we trained five models with distinct architectures and learning rules. Model 1 was a one-layer model with supervised and unsupervised learning

for the readout weights $\vec{v}$. Supervised learning corresponds to stochastic gradient descent on the binary cross-entropy as the loss function, which results in the learning rule

$$\Delta v_i = \eta(x_i(y - \hat{y}(x)) - 2\lambda v_i),$$

where $\eta$ is the learning rate, $\hat{y}$ is the model output, and $\lambda$ is the weight-decay parameter. Unsupervised learning corresponds to Hebbian learning with learning rule:

$$\Delta v_i = \eta(x_i(2\hat{y} - 1) - \lambda(2\hat{y} - 1)^2 v_i),$$

where, as before, $\eta$ is the learning rate and $\lambda$ is a weight-decay constant. In Model 1, we trained $\vec{v}$ using unsupervised learning during all passive exposures, and using both unsupervised learning and supervised learning during active trials.

The two-layer architecture introduces additional weights $W$ to map the input $\vec{x}$ to a latent representation $\vec{h} = W\vec{x}$, which is used by the readout weights $\vec{v}$ to produce the output $\hat{y} = S(\vec{v} \cdot W\vec{x})$. As for $\vec{v}$, the input weights $W$ can be trained by supervised learning and unsupervised learning. For supervised learning, we again used stochastic gradient descent on the binary cross-entropy, while, for unsupervised learning, Hebbian learning with weight decay was used:

$$\Delta W_{ij} = \eta \left( x_i h_j - \lambda \|\vec{h}\|^2 W_{ij} \right).$$

In Model 2, we trained $\vec{v}$ using unsupervised learning during all passive exposures and active trials, and $W$ using supervised learning during the active trials. In Model 3, we trained $W$ using unsupervised learning during all passive exposures and active trials, and $\vec{v}$ using supervised learning during the active trials. In Model 4, we expanded Model 3 by introducing an additional lateral set of weights $M$ between the hidden-layer neurons, which was trained during the passive exposures and active trials using anti-Hebbian learning:

$$\Delta M_{ij} = \eta(h_i h_j - \lambda M_{ij}).$$

Assuming that the neurons in the hidden layer quickly settle to a steady state, the effect of the lateral weights can be taken into account by using the hidden-layer representation at this equilibrium (*Pehlevan et al., 2015*):

$$\vec{h} = M^{-1}W\vec{x}.$$

In Model 5, we used the same architecture and learning rules as in Model 4, but additionally trained $W$ using supervised learning during active trials.

We chose the hyperparameters $\eta$ for all supervised learning rules such that the learning curve for the learners in the A-only schedule approximately matched the learning performance of the mice in the experiments. The learning rate for the unsupervised algorithms determines the separation of the learners with passive exposure and was chosen such that it maximized the separation while maintaining stability of the learning algorithm (the instability occurs when the learning rate for the unsupervised algorithms is set too high). The weight-decay parameters determine the asymptotic size of the weights when the learning algorithms converge. They were chosen such that the asymptotic norm of all weights matched the norm at initialization. The values for all hyperparameters can be found in *Table 1*.

## Acknowledgements

We thank Ciarra Thomas, Anna Easton, Komal Kaur, Gabriel Toea, and Brigid Deck for assistance with data collection.

This research was supported by the National Science Foundation (grant #2024926), the National Institutes of Health (grants R00NS114194 and R01NS118461), and the Office of the Vice President for Research & Innovation at the University of Oregon (through an I3 award).

## Additional information

### Funding

| Funder | Grant reference number | Author |
|--------|------------------------|--------|
| National Science Foundation | 2024926 | Melissa M Baese-Berk<br>Santiago Jaramillo |
| National Institute of Neurological Disorders and Stroke | Pathway to Independence Award R00NS114194 | James M Murray |
| National Institute of Neurological Disorders and Stroke | R01NS118461 | Santiago Jaramillo |
| University of Oregon | Office of the Vice President for Research & Innovation I3 award | Melissa M Baese-Berk<br>Santiago Jaramillo |

The funders had no role in study design, data collection, and interpretation, or the decision to submit the work for publication.

### Author contributions

Christian Schmid, Software, Formal analysis, Investigation, Methodology, Writing – original draft, Writing – review and editing; Muhammad Haziq, Investigation; Melissa M Baese-Berk, Conceptualization, Supervision, Funding acquisition, Writing – review and editing; James M Murray, Conceptualization, Formal analysis, Supervision, Funding acquisition, Investigation, Methodology, Writing – original draft, Project administration, Writing – review and editing; Santiago Jaramillo, Conceptualization, Data curation, Formal analysis, Supervision, Funding acquisition, Investigation, Methodology, Writing – original draft, Project administration, Writing – review and editing

### Author ORCIDs

Christian Schmid ⓘ http://orcid.org/0000-0002-3979-9169
James M Murray ⓘ https://orcid.org/0000-0003-3706-4895
Santiago Jaramillo ⓘ https://orcid.org/0000-0002-6595-8450

### Ethics

All procedures involving animals were carried out in accordance with National Institutes of Health Standards and were approved by the University of Oregon Institutional Animal Care and Use Committee (Protocol #21-26).

Reviewer #1 (Public Review): https://doi.org/10.7554/eLife.88406.3.sa1
Reviewer #2 (Public Review): https://doi.org/10.7554/eLife.88406.3.sa2
Reviewer #3 (Public Review): https://doi.org/10.7554/eLife.88406.3.sa3
Author Response https://doi.org/10.7554/eLife.88406.3.sa4

## Additional files

### Supplementary files
• MDAR checklist

### Data availability

The behavioral data for this study are available on Zenodo: https://doi.org/10.5281/zenodo.10360067. The code and output data for the computational modeling are available on github: https://github.com/cschmidat/behaviour-models (copy archived at *Schmid, 2023*).

The following dataset was generated:

| Author(s) | Year | Dataset title | Dataset URL | Database and Identifier |
|---|---|---|---|---|
| Schmid C, Haziq M, Baese-Berk M, Murray JM, Jaramillo S | 2023 | Behavioral data associated with "Passive exposure to task-relevant stimuli enhances categorization learning" | https://doi.org/ 10.5281/zenodo. 10360067 | Zenodo, 10.5281/ zenodo.10360067 |

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
