## [Editor Report · eLife assessment]

This study reports **valuable** behavioral and computational observations regarding how passive exposure to auditory stimuli can facilitate auditory categorization. The combination of behavioral results in mice with a study of artificial neural network models provides **solid** evidence for the authors' conclusions. This paper will likely be of broad interest to the general neuroscience community.

---

## [Referee Report · Reviewer #1 (Public Review)]

Schmid et al. investigate the question of how sensory learning in animals and artificial networks is driven both by passive exposure to the environment (unsupervised) and from reinforcing feedback (supervised) and how these two systems interact. They first demonstrate in mice that passive exposure to the same auditory stimuli used in a discrimination task modify learning and performance in the task. Based on this data, they then tested how the interaction of supervised and unsupervised learning in an artificial network could account for the behavioural results.

The clear behavioural impact of the passive exposure to sounds on accelerating learning is a major strength of the paper. Moreover, the observation that passive exposure had a positive impact on learning whether it was prior to the task or interleaved with learning sessions provides interesting constraints for modelling the interaction between supervised and unsupervised learning. A practical fallout for labs performing long training procedures is that the periods of active learning that require water-restriction could be reduced by using passive sessions. This could increase both experimental efficiency and animal well-being.

The modelling section clearly exhibits the differences between models and the step-by-step presentation building to the final model provides the reader with a lot of intuition about how supervised and unsupervised learning interact. In particular the authors highlight situations in which the task-relevant discrimination does not align with the directions of highest variance, thus reinforcing the relevance of their conclusions for the complex structure of sensory stimuli. A great strength of these models is that they generate clear predictions about how neural activity should evolve during the different training regimes that would be exciting to test.

As the authors acknowledge, the experimental design presented cannot clearly show that the effect of passive exposure was due to the specific exposure to task-relevant stimuli since there is no control group exposed to irrelevant stimuli. Studies have shown that exposure to a richer sensory environment, even in the adult, swiftly (ie within days) enhances responses even in the adult and even when the stimuli are different from those present in the task (1-3). Clearly distinguishing between these two options would require further experiments and could be a possible direction for future research.

1. Mandairon, N., Stack, C. & Linster, C. Olfactory enrichment improves the recognition of individual components in mixtures. Physiol. Behav. 89, 379-384 (2006).

2. Alwis, D. S. & Rajan, R. Environmental enrichment and the sensory brain: The role of enrichment in remediating brain injury. Front. Syst. Neurosci. 8, 1-20 (2014).

3. Polley, D. B., Kvašňák, E. & Frostig, R. D. Naturalistic experience transforms sensory maps in the adult cortex of caged animals. Nature 429, 67-71 (2004).

---

## [Referee Report · Reviewer #2 (Public Review)]

Schmid et al present a lovely study looking at the effect of passive auditory exposure on learning a categorization task.

The authors utilize a two-alternative choice task where mice have to discriminate between upward and downward moving frequency sweeps. Once mice learn to discriminate easy stimuli, the task is made psychometric and additional intermediate stimuli are introduced (as is standard in the literature). The authors introduce an additional two groups of animals, one that was passively exposed to the task stimuli before any behavioral shaping, and one that had passive exposure interleaved with learning. The major behavioral finding is that passive exposure to sounds improves learning speed. The authors show this in a number of ways through linear fits to the learning curves. Additionally, by breaking down performance based on the "extreme" vs "psychometric" stimuli, the authors show that passive exposure can influence responses to sounds that were not present during the initial training period. One limitation here is that the presented analysis is somewhat simplistic, does not include any detailed psychometric analysis (bias, lapse rates etc), and primarily focuses on learning speed. Ultimately though, the behavioral results are interesting and seem supported by the data.

To investigate the neural mechanisms that may underlie their behavioral findings, the authors turn to a family of artificial neural network models and evaluate the consequences of different learning algorithms and schedules, network architectures, and stimulus distributions, on the learning outcomes. The authors work through five different architectures that fail to recapitulate the primary behavior findings before settling on a final model, utilizing a combination of supervised and unsupervised learning, that was capable of reproducing the key aspects of the experiments. Ultimately, the behavioral results presented are consistent with network models that build latent representations of task-relevant features that are determined by statistical properties of the input distribution.

---

## [Referee Report · Reviewer #3 (Public Review)]

Summary of Author's Results/Intended Achievements

The authors were trying to ascertain the underlying learning mechanisms and network structure that could explain their primary experimental finding: passive exposure to a stimulus (independent of when the exposure occurs) can lead to improvements in active (supervised) learning. They modeled their task with 5 progressively more complex shallow neural networks classifying vectors drawn from multi-variate Gaussian distributions.

Account of Major Strengths:

Overall, the experimental findings were interesting. The modelling was also appropriate, with a solid attempt at matching the experimental condition to simplified network models.

---

## [Author Response]

The following is the authors’ response to the original reviews.

**Reviewer #1 (Public Review):**
The experimental design presented cannot clearly show that the effect of passive exposure was due to the specific exposure to task-relevant stimuli since there is no control group exposed to irrelevant stimuli.

We acknowledge the possibility that exposure to task-irrelevant stimuli could result in improvements in learning. Testing this possibility would be a worthwhile goal of future experiments, but it is outside the scope of our current study. We have been careful in our paper to only draw conclusions about the effects of exposure to task-relevant stimuli compared to no exposure. We have added a discussion of this point and relevant references to the literature in the Discussion section of our manuscript.

The conclusion that "passive exposure influences responses to sounds not used during training" (line 147) does not seem fully supported by the authors' analysis. The authors show that there is an increase in accuracy for intermediate sweep speeds despite the fact that this is the first time the animals encounter them in the active session. However, it seems impossible to exclude that this effect is not simply due to the increased accuracy of the extreme sounds that the animals had been trained on.

We have modified this sentence to emphasize that it refers to “intermediate” sounds. Regarding the reviewer’s concern, the conclusion is drawn from Figure 3, in which we show that mice exhibit an improvement on non-extreme stimuli after training on extreme stimuli. Panel 3D illustrates that the observed improvements are not just changes in psychometric performance driven by the extreme sounds. In the context of this result, the conclusion relates to generalization in performance on task-relevant stimuli that are closely related to the training stimuli. In our view, it was not entirely obvious a priori that this result would have to occur, since it is possible that performance could improve at the extremes without improving at the intermediate stimuli.

In the modelling section, the authors adjusted the hyper-parameters to maximize the difference between pure active and passive/active learning. This makes a comparison of learning rates between models somewhat confusing.

We apologize for the confusion. None of our conclusions are based on comparisons of learning speed between models, but perhaps this was not pointed out sufficiently clearly. The relevant comparisons between conditions for each specific model are made using the same hyperparameters. We have clarified this point in the modeling section of our manuscript.

The description of the sound does not state whether when reducing the slope of the sweeps the center or the onset frequency of the sounds is preserved.

Frequency modulated sounds of different FM slopes were generated such that the center frequency was always the same. This is now clarified in the updated version of the manuscript.

**Reviewer #1 (Recommendations for the authors):**
As mentioned, the specificity of the stimuli presented during the passive period is not explicitly addressed in either modelling or behaviour. For modelling, this could be quite straightforward to assess by manipulating the input stimuli during passive episodes. For the behaviour, this would require repeating the experiment with passive sessions during which unrelated sounds are presented (for example varying in frequency or intensity instead of frequency slope). I mainly include this suggestion to clarify my previous comment because this would require a huge amount of work.

We agree that varying the extent to which the presented passive stimuli are task-related to the task is an interesting point to study for future experiments. However, doing so for the experiments is outside the scope of the current study, and we believe exploring this only in the modeling part would add little value to the current study, because the outcome will highly depend on the details of the implementation.

**Reviewer #2 (Public Review):**
One limitation here is that the presented analysis is somewhat simplistic, does not include any detailed psychometric analysis (bias, lapse rates etc), and primarily focuses on learning speed.

In our preliminary analyses of trials that included extreme and intermediate stimuli after animals had learned the task (Figure 3), we investigated some metrics of the type that the reviewer suggests here. However, since such additional psychometric analyses were somewhat tangential to our main results (which are about learning speed and responses to sounds not included during training), we did not include these in our manuscript. In agreement with the reviewer’s concern, a main limitation of our study is that the available data does not allow for an analysis of psychometrics during the initial learning stages, since only the extreme stimuli were presented during the task.

**Reviewer #2 (Recommendations for the authors):**
The International Brain Lab has shown quite nicely that psychometric curves continue to improve (increased slope, decreased bias) across learning. This was not really discussed or presented in your data - is this observed during the S4 training portion?

We indeed saw improvements in the psychometric performance during stage S4, in particular for the active-only learners, as can be seen in Figure 3. We quantified these changes (now presented in the Results section), and added a discussion to the main text.

Why use a linear fit to extract the various quantities of interest? All of these quantities could be extracted from the raw behavioral data itself.

Because of the large variations in performance from day-to-day, a linear fit allowed us to extract a more reliable estimate of quantities like “Time to achieve 70%” and “Performance at 21 days” for each animal.

The analysis presented was focussed primarily on the fast learners. What about the slow learners? Are the ANN models able to recapitulate different aspects of their behavior?

We agree with the reviewer that the observation that the learners clustered into two groups calls for further investigation. In this study, we focused on the mice that learned more efficiently, because those allowed us to address our main research question about the influence of passive exposure. We believe, the slow learners could be modeled with ANNs that start with a less-easily discriminable input representation, which limits the performance that the trained network is ultimately able to achieve. This additional analysis is outside the scope of the current manuscript, but we hope to address these questions in the future.

Although I appreciate the thoroughness of the modeling, I was not entirely convinced by the narrative underlying models 1-5, since none of these models were able to successfully recapitulate your core findings. Would it not make more sense to focus primarily on the final model?

By starting with the simplest possible model that incorporates supervised and unsupervised learning, we were able to determine which ingredients were necessary to capture the behavioral data. We believe this could not have been clearly established by considering the final model alone.

**Reviewer #3 (Public Review):**
The first [major weakness] is that even Model 5 differs from their data. For example, the A+P (passive interleaved condition) learning curve in Figure 7 seems to be non-monotonic, and has some sort of complex eigenvalue in its decay to the steady state performance as trials increase. This wasn't present in their experimental data (Figure 2D), and implies a subtle but important difference. There also appear to be differences in how quickly the initial learning (during early trials) occurs for the A+P and A:P conditions. While both A+P and A:P conditions learn faster than A only in M5, A+P and A:P seem to learn in different ways, which isn't supported in their data.

The reviewer is correct that there are subtle differences between the two learning curves produced by Model 5. Due to expected variability in the experimental data, however, it is difficult to conclude whether such subtle distinctions also appear in the learning curves of the mice. Further, the slight overshoot of the learning curve that the reviewer mentions is not constrained by the experimental data due to different mice reaching asymptotic performance at different times, and many of them not having even reached asymptotic performance by the end of the training period.

However, even if there are minor discrepancies between the learning curves produced by the final version of the model and by the mice, we do not see this as being especially surprising or problematic. As in any model, there are a large number of potentially important features that are not included in any of our models–for example, realistic spectrotemporal neural responses, nonlinearity in neural activations, heterogeneity across mice, and many others. The aim of our modeling was to choose a space of possible models (which is inevitably restricted) and show which model version within that space best captures our experimental observations. Expanding the space of possible models that we considered to capture further nuances in the data will be a task for future work.

The second major weakness is that the authors also don't generate any predictions with M5.Can they test this model of learning somehow in follow-up behavioural experiments in mice? ... Without follow-up experiments to test their mechanism of why passive exposure helps in a schedule-independent way, the impact of this paper will be limited.

Although testing predictions from our models was beyond the scope of the current study, we do generate specific predictions with model M5 (in particular, about neural representations). Our model produces predictions about neural representations and the ways in which they evolve through learning, and we hope to test these predictions in future work.

I believe the authors need to place this work in the context of a large amount of existing literature on passive (unsupervised) and active (supervised) learning interactions. This field is broad both experimentally and computationally. For example, there is an entire sub-field of machine learning, called semi-supervised learning that is not mentioned at all in this work.

We thank the reviewer for pointing this out. The Discussion section of the updated manuscript now includes a discussion on how our results fit in with this literature.

**Reviewer #3 (Recommendations for the authors):**

All points made by the reviewer in their Recommendations For The Authors are associated with those presented in the Public Review and they are addressed in our response above.